# Nature and Clinical Outcomes of Acute Hemorrhagic Rectal Ulcer

**DOI:** 10.3390/diagnostics12102487

**Published:** 2022-10-14

**Authors:** Yasutaka Takahashi, Yosuke Shimodaira, Tamotsu Matsuhashi, Tsuyotoshi Tsuji, Sho Fukuda, Kae Sugawara, Youhei Saruta, Kenta Watanabe, Katsunori Iijima

**Affiliations:** 1Department of Gastroenterology, Akita University Graduate School of Medicine, Akita 010-8543, Japan; 2Department of Gastroenterology, Akita City Hospital, Akita 010-0933, Japan

**Keywords:** acute hemorrhagic rectal ulcer, lower gastrointestinal bleeding, in-hospital mortality

## Abstract

Acute hemorrhagic rectal ulcer (AHRU) is a relatively rare disease that can lead to massive hematochezia. Although AHRU is a potentially life-threatening disease, its characteristics and clinical course are not fully understood. In this study, the clinical features were compared between AHRU and lower gastrointestinal bleeding (LGIB) from other causes (non-AHRU). Then, risk factors for all-cause in-hospital mortality in patients with AHRU were identified. A total of 387 consecutive adult patients with LGIB who were managed at two tertiary academic hospitals in Akita prefecture in Japan were retrospectively enrolled. Subjects were divided into AHRU and non-AHRU groups according to the source of bleeding. Regression analyses were used to investigate significant associations, and the results were expressed as odds ratios (ORs) and 95% confidence intervals (CIs). AHRU was found as the bleeding source in 72 (18.6%) of the patients. In comparison to non-AHRU, having AHRU was significantly associated with in-hospital onset, age > 65 years, and systolic blood pressure < 90 mmHg. The AHRU group had a significantly higher in-hospital mortality rate in comparison to the non-AHRU group (18.0% vs. 8.3, *p* = 0.02), and hypoalbuminemia (<2.5 g/dL) was significantly associated with in-hospital mortality in the AHRU group (OR, 4.04; 95%CI, 1.11–14.9; *p* = 0.03). AHRU accounts for a substantial portion (18.6%) of LGIB in our area, where the aging rate is the highest in Japan. Since AHRU is a potentially life-threatening disease that requires urgent identification and management, further studies to identify robust risk factors associated with serious clinical outcomes are required.

## 1. Introduction

Acute hemorrhagic rectal ulcer (AHRU) is a relatively rare disease that is responsible for massive hematochezia. Soeno et al. first described the term “AHRU” in a report from our institute (Akita University Hospital) in Japan in 1981 [1]. They described four cases who showed cerebral ischemia as the acute onset of painless and massive rectal bleeding [1]. Since then, substantial numbers of case series and clinical studies have been reported in Asian countries [2,3,4,5,6,7,8,9,10,11,12,13], and more recently, some reports have indicated that AHRU also exists in Western countries [14,15,16,17]. Currently, AHRU is recognized to be characterized by sudden onset, painless, massive, and fresh rectal bleeding, which frequently occurs in elderly patients with serious complications [6,9,12].

With an aging population, the incidence of AHRU seems to be increasing in Japan. Although previous studies from other areas reported that AHRU accounted for 2.8–8% of the causes of lower gastrointestinal bleeding (LGIB) [2,6,15], we have recently reported that AHRU accounts for 18.6% of LGIB in Akita prefecture in Japan [18], which is the region with the fastest-aging population in the country [19,20]. A similar phenomenon will be reproduced in other parts of Japan and other countries that are facing an aging society. However, its characteristics and clinical course are not fully understood.

Previously, the clinical course of AHRU was considered to be favorable once hemostasis was achieved [3,14]. Subsequent studies, however, indicated that it is not necessarily favorable due to patients’ underlying serious complications [7,8,11], although few properly compared the clinical course between AHRU and LGIB from other causes [7]. In addition, so far, only a few demonstrated risk factors for serious clinical outcomes, such as rebleeding [6,8,11] and mortality [7], have been observed among patients with AHRU. Intensive medical care based on the risk factors associated with such worse-case clinical outcomes would be necessary to improve the clinical course of AHRU.

We recently collected clinical data on LGIB in our area and reported the usefulness of various scoring systems to predict serious clinical outcomes, such as in-hospital mortality [18]. Using this database, we initially compared clinical features and treatment outcomes of AHRU with those of LGIB from other causes (non-AHRU). Then, we attempted to identify risk factors for rebleeding and in-hospital mortality among patients with AHRU.

## 2. Methods

### 2.1. Patients

In our original study [18], we retrospectively enrolled a total of 387 consecutive adult patients who were hospitalized with LGIB over 6 years at two tertiary academic hospitals in Akita prefecture in Japan (Akita City Hospital and Akita University Hospital) from 2015 to 2020. Those with mild LGIB who did not require hospitalization were excluded since the main outcome of the original study was in-hospital mortality [18]. Both those with hospitalization for LGIB (out-of-hospital onset) or the development of LGIB during hospitalization for another indication (in-hospital onset) were included in the study.

A colonoscopy was performed emergently or electively with or without computed tomography (CT) in all patients with LGIB in order to identify the bleeding source. Then, depending on the source of bleeding, the entire LGIB cohort was divided into two groups: the AHRU group and the (non-AHRU) group. Both definite and presumptive sources of LGIB were included. Definite sources of bleeding were defined as lesions with documented visualization of active bleeding, a visible vessel, or adherent clot (stigmata of recent hemorrhage). Presumptive diagnoses were defined as cases of diverticula, hemorrhoids, or angiodysplasia without stigmata of recent bleeding.

This study protocol was reviewed and approved by the ethics committees of the two participating institutes (2676).

### 2.2. Data Collection

The following patient’s medical information and data were collected from electronic medical records: patient demographics (age and sex); in-hospital or out-of-hospital onset; altered mental status; vital signs (systolic blood pressure and pulse); physical condition (Eastern Cooperative Oncology Group Performance Status: ECOG-PS); comorbid conditions (Charlson Comorbidity Index: CCI); blood test results (hemoglobin, albumin, and creatinine; international normalized ratio (INR); and blood urea nitrogen (BUN)); and medication (anticoagulants, antiplatelet agents, nonsteroidal anti-inflammatory drugs, and steroids) at the onset of LGIB. In addition, additional data required to determine the scores of 3 scoring systems (CHAMPS, NOBLADS, and ABC) [19,21,22,23], which were validated to predict the serious clinical outcomes following LGIB, were also collected (e.g., abdominal symptoms or stool condition).

### 2.3. Definitions

LGIB was defined as the presentation of hematochezia, including red blood or clots per rectum, maroon-colored stool, or blood mixed in with stool, without any findings indicative of upper gastrointestinal bleeding at esophagogastroduodenoscopy. AHRU was defined as acute and painless hematochezia with endoscopic documentation of rectal ulcer(s) [1]. In-hospital mortality was defined as death during the index hospitalization, whatever the cause [24]. Re-bleeding was suspected by the presence of fresh hematochezia and circulating instability after successful hemostasis and was defined as a new bleeding episode from the same source based on an endoscopic examination.

### 2.4. Statistics

Continuous variables were expressed as the median and interquartile range and were compared using the Mann–Whitney U test. Categorical valuables were expressed as the number and proportion, and were compared using Fisher’s exact test or the chi-square test, as appropriate.

Initially, clinical features and treatment outcomes were compared between AHRU and non-AHRU, and factors associated with having AHRU (relative to non-AHRU) were identified. Then, we attempted to identify factors associated with rebleeding and in-hospital mortality in AHRU patients. In these regression analyses, factors that showed a *p* value of <0.20 in univariate analysis were included in multivariate analysis, and the results were expressed as odds ratios (ORs) and 95% confidence intervals (CIs).

The diagnostic performance of 3 scoring systems for predicting rebleeding and in-hospital mortality among patients with AHRU was assessed by a receiver operating characteristic curve analysis, and the area under the receiver operating characteristic curve (AUC) was calculated. All analyses were conducted using the EZR software program (Saitama Medical Center, Jichi Medical University, Saitama, Japan) [25]. *p* value of <0.05 were considered statistically significant.

## 3. Results

Based on colonoscopy and CT examination, among 387 enrolled patients with major LGIB, AHRU was found as the bleeding source in 72 (18.6%) cases, as was reported in our recent study [18]. These patients constituted the AHRU group. The remaining 316 patients (diverticular bleeding, *n* = 132; ischemic colitis, *n* = 54; delayed post-polypectomy-induced bleeding, *n* = 23; hemorrhoid bleeding, *n* = 21; bleeding colonic cancer, *n* = 18; bleeding colitis, *n* = 13; telangiectasia, *n* = 8; and unknown/others, *n* = 32) constituted the non-AHRU group. Representative endoscopic images of AHRU are shown in Figure 1.

AHRU mainly occurs in elderly individuals (median age: 79 years) with several comorbidities (median CCI: of 3), and 50% (36 of 72) of cases involve bedridden patients (ECOG-PS = 3 or 4) as previously reported [6,9,12]. Table 1 shows the clinical factors in the AHRU and non-AHRU groups; there were significant differences in various factors between the two groups.

In particular, in comparison to the non-AHRU group, the AHRU group showed higher age, a poorer general condition indicated by a higher ECOG-PS score, higher CCI score, and higher proportion of in-hospital onset, more massive bleeding indicated by lower blood pressure and lower hemoglobin, and a poorer nutritional status indicated by a lower serum albumin level. On the other hand, there were no significant differences in sex or medication between the two groups. Consequently, the AHRU group showed higher scores in all three scoring systems, with a highly significant difference (all *p* < 0.0001) in comparison to the non-AHRU group (CHAMPS: 2.5 (2, 4) vs. 1 (0, 2); NOBLADS: 4 (3, 5) vs. 3 (2, 4); ABC: 3.5 (2, 7) vs. 2 (1, 4)), suggesting that AHRU was potentially associated with worse clinical outcomes.

Characteristics of AHRU in relation to non-AHRU were investigated using logistic regression analyses (Table 2).

Various factors, including higher ECOG-PS, CCI, in-hospital onset, and hypoalbuminemia, showed a highly significant association with AHRU in univariate analysis. However, the statistical significance of many of these factors was lost in multivariate analysis, which left only three factors associated with AHRU. In-hospital onset showed the strongest association with AHRU (OR, 9.65; 95%CI, 2.60–20.6), followed by age >65 years (OR, 4.03; 95%CI, 1.53–10.7), and systolic blood pressure <90 mmHg (OR, 4.01; 95%CI, 1.65–10.6).

Comparisons of the treatment outcomes between the AHRU and non-AHRU groups are shown in Table 3.

The AHRU group more frequently received both endoscopic hemostasis and blood transfusion in comparison to the non-AHRU group (68% vs. 26%, *p* < 0.0001 for hemostasis; 41.6% vs. 23.2%, *p* = 0.003 for blood transfusion), which reflected the observation that the AHRU group had more massive bleeding in comparison to the non-AHRU group (Table 1 and Table 2). There was no significant difference in rebleeding rates between the two groups, in either the entire cohort or in the specific cohort that received endoscopic hemostasis. Noticeably, there was a significant difference in the in-hospital mortality rates between the two groups; namely, the in-hospital mortality rate was 18.0% in the AHRU group, which was more than double the rate in the non-AHRU group (8.3%, *p* = 0.027). The causes of 13 deaths in the AHRU group were malignancy (*n* = 5), pneumonia (*n* = 2), sepsis (*n* = 2), heart failure (*n* = 1), renal failure (*n* = 1), respiratory failure (*n* = 1), and senility (*n* = 1); thus, no one died directly from uncontrolled bleeding.

The diagnostic performance of the three scoring systems (CHAMPS, NOBLADS, and ABC) to predict rebleeding and in-hospital mortality among patients with AHRUs is shown in Table 4. 

Overall, the performance of these scoring systems is largely suboptimal, especially for the prediction of in-hospital mortality. For instance, although we recently developed the CHAMPS score and demonstrated good performance of the score to predict in-hospital mortality in the entire LGIB cohort with an AUC (95%CI) of 0.80 (0.73–0.87) [18], the performance turned out to be rather poor in the sub-analysis in which the population was restricted to patients with AHRU (AUC, 0.57; 95%CI, 0.39–0.76).

After demonstrating the poor performance of the three existing scoring systems in the prediction of serious clinical outcomes within the AHRU group, we investigated the associations of individual clinical factors with rebleeding or in-hospital mortality in this group (Table 5).

Although some factors were significantly associated with rebleeding in univariate analysis (e.g., systolic blood pressure < 90 mmHg, hemoglobin < 10.0 g/dL, serum albumin < 2.5 g/dL, and steroid intake), the statistical significance of these factors was lost in multivariate analysis. On the other hand, serum albumin level of <2.5 g/dL was a single, statistically significant factor associated with in-hospital mortality in univariate analysis (OR, 3.70; 95%CI, 1.05–13.1; *p* = 0.04) and the statistical significance remained in multivariate analysis (OR, 4.04; 95%CI, 1.11–14.9; *p* = 0.03).

## 4. Discussion

Using the database of a consecutive LGIB cohort, we found several important findings regarding the characteristics and treatment outcomes of AHRU, a relatively rare disease of unknown nature. AHRU, which accounted for 18.6% of LGIB in our cohort, was characterized by older age, in-hospital onset of bleeding, and massive bleeding, reflected by systolic blood pressure < 90 mmHg. Importantly, the AHRU group showed higher in-hospital mortality in comparison to the non-AHRU group, although no patients died directly from bleeding. Further, the in-hospital mortality was associated with hypoalbuminemia in AHRU patients; however, it is still challenging to predict by existing scoring systems.

Previous studies consistently reported that old age, poor performance status (e.g., bedridden patients), severe comorbidity, and hypoalbuminemia were significantly associated with the onset of AHRU [8,9,10,12]. Our univariate analysis identified several factors associated with AHRU, including older age, higher ECOG-PS, higher CCI, and hypoalbuminemia. Nonetheless, in multivariate analysis, in-hospital onset, age > 65 years, and systolic blood pressure <90 mmHg were the only factors associated with AHRU, and other factors lost their significant associations. The onset of LGIB during hospitalization for other diseases (in-hospital onset) was a compound factor that included various factors associated with the patient’s general conditions (e.g., performance status, comorbidities, and nutrition status), which made in-hospital onset the strongest factor associated with AHRU and made other related-factors insignificant in multivariate analysis.

Previous studies demonstrated high rates of mortality in AHRU, ranging from approximately 15% to 25% [2,4,9,10,11,15,16], although none had appropriate comparative controls. Consistently, the in-hospital mortality rate in our AHRU group was 18.0%, and our study extended these previous studies by clearly showing that in-hospital mortality was much higher in comparison to an appropriate control (the non-AHRU group). Nonetheless, no patients in the AHRU group died directly from uncontrolled bleeding, which was consistent with some previous studies which showed that the majority of patients with AHRU died from causes unrelated to bleeding [2,6,7,9,10,11,16]. This reinforces the importance of systemic care after hemostasis rather than merely focusing on local hemostasis, especially in AHRU, which is likely to occur in individuals with a poor general condition and serious comorbidities [24].

Several scoring systems have been developed to predict serious clinical outcomes following gastrointestinal bleeding, and previous studies have successfully demonstrated the usefulness of some scoring systems not only for upper gastrointestinal bleeding but also for LGIB [18,22,23]. Nonetheless, the existing scoring systems failed to show a significant ability to discriminate between survivors and non-survivors in AHRU, which represents a more severe subset of LGIB. Hence, once physicians find that the origin of hematochezia is AHRU, there is no scoring system to predict the clinical course after initial hemostasis.

Accordingly, we attempted to identify factors associated with rebleeding and mortality in AHRU. Thus far, three studies have investigated factors associated with rebleeding in AHRU [6,8,11]. These studies reported that CCI score, severity of comorbidities, abnormal coagulation, and ulcer morphology (e.g., whole circumferential ulcer) were risk factors for rebleeding [6,8,11]. Consistently, we also found that some of these factors were significantly associated with rebleeding in univariate analysis of patients with AHRU. However, we failed to identify any significant factors in multivariate analysis, probably due to insufficient statistical power as the study population was relatively small.

On the other hand, a single previous study investigated factors associated with mortality among 36 patients with AHRU occurring after ICU admission, and found that thrombocytopenia was a risk factor for 4-week mortality [7]. In the present study, we identified hypoalbuminemia as a single significant risk factor for in-hospital mortality among 72 patients with AHRU in both univariate and multivariate analyses. As a reference, we collected additional data on platelet count in patients with AHRU, but we did not find any significant differences between non-survivors and survivors (e.g., median platelet count (interquartile range): 190,000/µL (110,000, 230,000) vs. 200,000/µL (160,000, 570,000), *p* = 0.49; percentage of thrombocytopenia defined as <100,000/µL: 23% vs. 10.2%, *p* = 0.35). Interestingly, although all three scoring systems investigated in this study (CHAMPS, NOBLADS, and ABC) included hypoalbuminemia as a variable [18,21,22,23], they failed to show good diagnostic performance in the prediction of in-hospital mortality in AHRU. Thus, the scores need to be modified to satisfactorily predict clinical outcomes in AHRU.

Regarding the etiology of AHRU, Nakamura et al. reported that rectal mucosal blood flow was reduced in the supine position during bed rest and that the reduction in blood flow could be responsible for the development of AHRU, which frequently occurs in bedridden patients [26]. However, a substantial portion of AHRU occurs in non-bedridden individuals [6], and this was true for the current study, in which only 50% of the patients were bedridden. Subsequently, Motomura et al. proposed that hypoalbuminemia suggestive of malnutrition was another main contributor to the development of AHRU because AHRU patients universally showed hypoalbuminemia [6,14]. Malnutrition could predispose a patient to intestinal mucosal damage, at least by delaying healing process of the injury [27,28]. We also found that hypoalbuminemia was involved in various clinical aspects of AHRU (e.g., it was not only associated with the onset of AHRU but also with the treatment outcomes), suggesting that it should play a pivotal role in the pathogenesis of AHRU. Thus, nutritional support may be important not only for the prevention of AHRU but also for improving its clinical course.

One limitation of this study was the relatively small number of AHRU patients (*n* = 72), which precluded solid analyses to identify factors associated with the clinical outcomes. However, AHRU is still a rare disease, and we enrolled a relatively large number of patients with AHRU in comparison to most previous studies [1,2,3,4,5,7,8,9,10,12,13,14,15,16,17] by enrolling consecutive LGIB patients from two tertiary medical centers in Akita prefecture in Japan (an AHRU-prone area due to the aging population) over a 6-year period.

## 5. Conclusions

In conclusion, AHRU accounts for a substantial portion (18.6%) of major LGIB in Akita prefecture, where the aging rate is the highest in Japan, and this phenomenon can be expected to be observed in other areas in Japan and other countries with an aging population. Since AHRU is a potentially life-threatening disease that requires urgent identification and management, further studies are required to identify robust risk factors associated with clinical outcomes.

## Figures and Tables

**Figure 1 diagnostics-12-02487-f001:**
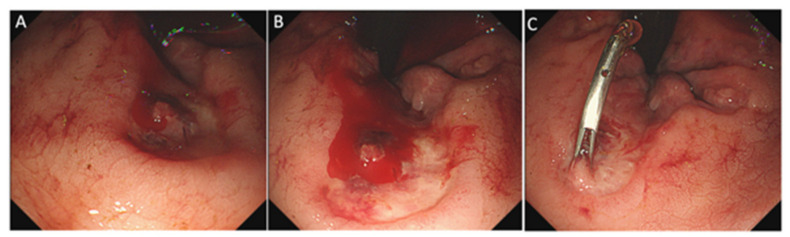
Endoscopic image of an acute hemorrhagic rectal ulcer. Colonoscopy revealed a solitary round rectal ulcer adjacent to the dentate line (**A**) and active bleeding with a stream of blood coming from an exposed vessel at the bottom of the ulcer (**B**). Endoscopic hemostasis was successfully achieved with clipping (**C**).

**Table 1 diagnostics-12-02487-t001:** Comparisons of clinical features between AHRU and non-AHRU lower gastrointestinal bleeding.

	AHRUs (*n* = 72)	Non-AHRUs (*n* = 315)	*p* Value
Demographics			
Age: years	79 (70.75, 85)	74 (63, 81)	0.0003
Sex: male: *n*	36 (50%)	186 (59%)	0.15
ECOG-PS: score	2 (1, 4)	1 (0, 2)	<0.0001
CCI: score	3 (1, 4.25)	1 (0, 3)	<0.0001
In-hospital onset of bleeding: *n*	50 (69.4%)	56 (17.7%)	<0.0001
Systolic blood pressure: mmHg	119 (97.5, 134.75)	126 (108, 142)	0.008
Heart rate: bpm	85 (76, 100)	80 (70, 93)	0.005
Altered mental status: *n*	9 (12.5%)	11 (3.4%)	0.005
Blood test			
Hemoglobin: g/dL	10.05 (8.3, 11.725)	11.2 (11.2, 13.3)	0.015
Serum Albumin: g/dL	2.8 (2.3, 3.2)	3.6 (3.1, 4.1)	<0.0001
BUN: mg/dL	24.2 (15.4, 34.9)	18.2 (13.4, 27.9)	0.003
Serum Creatinine	0.8 (0.6, 1.3)	0.8 (0.6, 1.1)	0.59
INR	1.1 (1, 1.4)	1.0 (1, 1.2)	0.021
Medication			
NSAIDs: *n*	10 (13.8%)	43 (13.7%)	1.000
Antithrombotics: *n*	35 (48.6%)	134 (42.5%)	0.43
Steroid: *n*	10 (13.8%)	26 (8.2%)	0.16
Complication			
Diabetes mellitus: *n*	21 (29.1%)	56 (17.8%)	0.034
Hypertension: *n*	39 (54.1%)	150 (47.6%)	0.36
Hyperlipidemia: *n*	17 (23.6%)	69 (21.9%)	0.76
Scoring			
CHAMPS score	2.5 (2, 4)	1 (0, 2)	<0.0001
NOBLADS score	4 (3, 5)	3 (2, 4)	<0.0001
ABC score	3.5 (2, 7)	2 (1, 4)	<0.0001

Continuous values and categorical values are expressed as the median (interquartile range) and *n* (%), respectively. AHRUs, acute hemorrhagic rectal ulcers; ECOG-PS, Eastern Cooperative Oncology Group performance status; CCI, Charlson comorbidity index; BUN, Blood urea nitrogen; INR, International normalized ratio; NSAIDs, nonsteroidal anti-inflammatory drugs.

**Table 2 diagnostics-12-02487-t002:** Factors associated with AHRU among patients with lower gastrointestinal bleeding.

Factors	References	OR (95% CI), *p* Value	
		Univariate	Multivariate
**Age: >65 years**	≤65 years	**2.42 (1.19–4.94), 0.015**	**4.03 (1.53–10.7), 0.005**
Sex: male	female	0.68 (0.41–1.14), 0.11	0.78 (0.39–1.55), 0.48
ECOG-PS: ≥2	<2	**3.67 (2.17–6.2), <0.0001**	1.62 (0.81–3.24), 0.17
CCI: ≥2	<2	**2.63 (1.56–4.43), 0.0003**	0.90 (0.42–1.94), 0.79
In-hospital onset of GI bleeding: yes	no	**10.4 (5.85–18.6), <0.0001**	**9.65 (4.60–20.6), <0.0001**
Systolic blood pressure: <90 mmHg	≥90 mmHg	**3.82 (1.84–7.90), 0.0003**	**4.01 (1.65–10.6), 0.005**
Heart rate: >100 bpm	≤100 bpm	1.77 (0.95–3.32), 0.073	2.07 (0.90–4.77), 0.088
Altered mental status: yes	no	**3.92 (1.56–9.86), 0.0037**	3.27 0.94–11.3), 0.062
Hemoglobin: <10.0 g/dL	≥10.0 g/dL	**1.75 (1.04–2.93), 0.035**	0.67 (0.30–1.48), 0.32
Serum albumin: <2.5 g/dL	≥2.5 g/dL	**5.86 (3.10–11.10), <0.0001**	2.07 (0.81–4.98), 0.12
BUN: >20 mg/dl	≤20 mg/dl	**1.86 (1.11–3.15), <0.0001**	1.78 (0.90–3.53), 0.10
INR: >1.5	≤1.5	1.41 (0.70–2.84), 0.37	-
NSAIDs: yes	no	0.85 (0.41–1.77), 0.66	-
Antithrombotic drugs: yes	no	1.26 (0.76–2.11), 0.37	-
Steroid: yes	no	1.78 (0.82–3.88), 0.15	1.28 (0.43–3.82), 0.66

Bold typeface represents <0.05. AHRU, acute hemorrhagic rectal ulcer; OR, odds ratio; CI, confidence interval; ECOG-PS, Eastern Cooperative Oncology Group performance status; CCI, Charlson comorbidity index; BUN, Blood urea nitrogen; INR, International normalized ratio; NSAIDs, nonsteroidal anti-inflammatory drugs.

**Table 3 diagnostics-12-02487-t003:** Treatment outcomes of AHRU in comparison to non-AHRU lower gastrointestinal bleeding.

	AHRUs (*n* = 72)	Non-AHRUs (*n* = 315)	*p* Value
⮚ Stigmata of recent hemorrhage	30 (41.6%)	110 (34.9%)	0.34
⮚ Endoscopic hemostasis	49 (68%)	82 (26%)	<0.0001
⮚ Patients who received blood transfusion	30 (41.6%)	73 (23.2%)	0.003
⮚ Rebleeding	11 (15.2%)	61 (19.4%)	0.86
⮚ Rebleeding after endoscopic hemostasis	10 (20.4%)	20 (24.4%)	0.67
⮚ In-hospital mortality	13 (18%)	26 (8.3%)	0.027

Data are expressed as *n* (%). AHRUs, acute hemorrhagic rectal ulcers.

**Table 4 diagnostics-12-02487-t004:** Diagnostic performance of 3 scoring systems to predict rebleeding and in-hospital mortality among patients with AHRUs.

Scoring	Rebleeding				In-Hospital Mortality			
	With	Without	*p* Value	AUC (95% CI)	With	Without	*p* Value	AUC (95% CI)
CHAMPS	4 (2.5, 4)	2 (2, 3)	0.08	0.66 (0.49–0.84)	3 (2, 4)	2 (2, 4)	0.40	0.57 (0.39–0.76)
NOBLADS	4 (3, 4)	4 (4, 5)	0.04	0.69 (0.55–0.83)	4 (4, 5)	4 (3, 5)	0.24	0.60 (0.44–0.76)
ABC	2 (2, 5)	4 (2, 7)	0.30	0.60 (0.42–0.77)	5 (2, 7)	3 (2, 7)	0.36	0.58 (0.43–0.73)

Scoring values are expressed as the median (interquartile range). AHRUs, acute hemorrhagic rectal ulcers AUC, area under the receiver operating characteristic curve; CI, confidence interval.

**Table 5 diagnostics-12-02487-t005:** Factors associated with re-bleeding and in-hospital mortality among patients with AHRUs.

Factors	Reference	Rebleeding		In-Hospital Mortality	
		Univariate CI (95% CI), *p* Value	Multivariate CI (95% CI), *p* Value	Univariate CI (95% CI), *p* Value	Multivariate CI (95% CI), *p* Value
**Age: >65 years**	≤65 years	0.40 (0.09–1.84), 0.24	-	0.99 (0.19–5.24), 0.99	-
Sex: male	female	0.81 (0.22–2.93), 0.74	-	2.67 (0.74–9.63), 0.13	2.68 (0.69–10.4), 0.15
ECOG-PS: ≥2	Score = 0 or 1	1.06 (0.279–4.02), 0.93	-	0.64 (0.19–2.17), 0.48	-
CCI: ≥2	Score = 0 or 1	0.42 (0.11–1.60), 0.21	-	2.82 (0.57–14.0), 0.20	-
In-hospital onset of bleeding: yes	no	5.25 (0.63–43.8), 0.13	4.09 (0.41–40.6), 0.23	0.99 (0.27–3.63), 0.98	-
Systolic blood pressure: <90 mmHg	≥90 mmHg	**4.25 (1.08–16.7), 0.04**	3.28 (0.68–15.8), 0.14	1.94 (0.50–7.46), 0.33	-
Hemoglobin: <10.0 g/dL	≥10.0 g/dL	**6.48 (1.29–32.6), 0.02**	4.50 (0.59–34.7), 0.15	1.38 (0.41–4.61), 0.60	-
Serum albumin: <2.5 g/dL	≥2.5 g/dL	**3.91 (1.01–15.2), 0.049**	1.14 (0.19–6.99), 0.89	**3.70 (1.05–13.1), 0.04**	**4.07 (1.11–14.9), 0.03**
INR: >1.5	≤1.5	0.38 (0.04–3.30), 0.38	-	0.26 (0.03–2.53), 0.27	-
Antithrombotic drugs: yes	no	0.55 (0.15–2.08), 0.38	-	0.60 (0.18–4.01), 0.42	-
Steroid: yes	no	**5.24 (1.18–23.2), 0.029**	3.88 (0.70–21.5), 0.12	0.46 (0.053–4.01), 0.48	-
Rebleeding: yes	no	-	-	1.01 (0.19–5.34), 0.27	-

Bold typeface represents <0.05. AHRUs, acute hemorrhagic rectal ulcers; OR, odds ratio; CI, confidence interval; ECOG-PS, Eastern Cooperative Oncology Group performance status; CCI, Charlson comorbidity index; INR, International normalized ratio.

## Data Availability

The data presented in this study are available on request from the corresponding author. The data are not publicly available due to ethical.

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
