# Peer review of "Nature and Clinical Outcomes of Acute Hemorrhagic Rectal Ulcer"

_diagnostics, 2022, doi:10.3390/diagnostics12102487_

Round 1
Reviewer 1 Report
Using some of these update articles as Ref(s). could be nice:
Okamoto T, Takasu A, Yoshimoto T, Yamamoto K, Shiratori Y, Ikeya T, Fukuda K
Digital compression for hemostasis in acute hemorrhagic rectal ulcer: a report of 4 cases and review of the literature.
Clinical journal of gastroenterology. 2021
Nadi A, Cherouaqi Y, Oulammou Z, Delsa H, Rouibaa F
The Impact of Argon Plasma Coagulation in the Treatment of a Solitary Rectal Ulcer Syndrome Revealed by Massive Hemorrhage.
Cureus. 2022
Iqbal U, Ahmed Z, Anwar H, Shah NM, Lee W, Nawras A, Khara HS, Ahmed A, Khurana S
Hemorrhagic Ascites Is Associated With Reduced Survival in Cirrhosis: A Systematic Review and Meta-Analysis.
Gastroenterology research. 2022
Nagata N, Kobayashi K, Yamauchi A, Yamada A, Omori J, Ikeya T, Aoyama T, Tominaga N, Sato Y, Kishino T, Ishii N, Sawada T, Murata M, Takao A, Mizukami K, Kinjo K, Fujimori S, Uotani T, Fujita M, Sato H, Suzuki S, Narasaka T, Hayasaka J, Funabiki T, Kinjo Y, Mizuki A, Kiyotoki S, Mikami T, Gushima R, Fujii H, Fuyuno Y, Gunji N, Toya Y, Narimatsu K, Manabe N, Nagaike K, Kinjo T, Sumida Y, Funakoshi S, Kawagishi K, Matsuhashi T, Komaki Y, Miki K, Watanabe K, Fukuzawa M, Itoi T, Uemura N, Kawai T, Kaise M
Identifying Bleeding Etiologies by Endoscopy Affected Outcomes in 10,342 Cases With Hematochezia: CODE BLUE-J Study.
The American journal of gastroenterology. 2021
Benevides ML, Elias S, Fagundes DA, Martins RF, Dutra MM, Rodrigues de Oliveira Thais ME, Rodrigues GM, Nunes JC, Martins GL
Acute Hemorrhagic Leukoencephalopathy Triggered by COVID-19 Infection.
The Neurohospitalist. 2022
Reviewer 2 Report
The present work was designed to the clinical features of acute hemorrhagic rectal ulcer (AHRU) and risk factors for all-cause in-hospital mortality in patients with AHRU. However, some questions should be clarified before the paper published.
1. The author should label the manuscript with line numbers for easy positioning.
2. Line 5, Please note that the format of the references is uniform in the introduction.
3. Line 8 in introduction, “…massive, fresh…” should be changed to “…massive, and fresh…”.
4. Line 11 in introduction, “For instance” should be deleted.
5. “P values of <0.05 were considered to indicate statistical significance”, Please rewritten this sentence.
6. Line 5 in results, “…bleeding n=23” should be changed to “…bleeding, n=23”.
7. Lines 12-16, please explained this sentence.
8. In discussion, there are mainly repetition of results.
9. There are also various typo and grammar errors that authors should correct before further process of the manuscript.
Reviewer 3 Report
In the manuscript submitted by Takahashi et al, the authors conducted a retrospective study to evaluate the clinical features of acute hemorrhagic rectal ulcer (AHRU) and identify possible risk factors for all-cause in-hospital mortality of AHRU patients. As a rare but potentially life-threatening disease, the purpose of this study is of importance. Nonetheless, although the authors did show certain aspects that are associated with AHRU, and identified hypoalbuminemia as a risk factor for in-hospital mortality, these indicators are not very specific. Since the authors pointed out the limited number of patients probably accounted for these relatively ambiguous factors, it is suggested that the authors should also conduct a meta-analysis with published literatures, which might improve the quality of the findings, or at least support the current finding.
Round 2
Reviewer 2 Report
The authors have revised their manuscript accordingly.
Reviewer 3 Report
No further question.